# Reliability Modeling of Products with Self-Recovery Features for Competing Failure Processes in Whole Life Cycle

**Xingang Wang [1,2,3], Baoyan Wang [1,*], Yiqun Niu [4] and Ziqiu He [1]**

1   College of Mechanical and Electrical Engineering, Guangdong University of Petrochemical Technology, Maoming 525000, China
2   State Key Laboratory of Precision Electronic Manufacturing Technology and Equipment, Guangzhou 510006, China
3   School of Mechanical and Electronic Engineering, Quanzhou University of Information Engineering, Quanzhou 362000, China
4   School of Mechanical Engineering, Liaoning Petrochemical University, Fushun 113001, China
*   Correspondence: xgwang@neuq.edu.cn; Tel.: +86-191-2090-7010

**Abstract:** In light of the increasing demand for the reliability analysis of self-recovery products, with features of limited storage period and multistage degradation, a reliability evaluation model in whole life cycle is proposed. The degradation process comprises one storage phase and two working phases. On the basis of the idea of competitive failure, the shock process and the feature of self-recovery were introduced into the model. Furthermore, the problem of it being difficult to add variables to a reliability model is solved with the use of the Stieltjes integral. The influences of the parameters of model reliability are analyzed, and the results demonstrate that the new model could adequately describe the competing failure process. The model also exhibited certain feasibility and theoretical reference values.

**Keywords:** self-recovery; multistage degradation; whole life cycle; competing failure





## 1. Introduction

In traditional reliability analysis, the product life is considered the main research object, and the evaluation results need large amounts of supporting failure data; this method is suitable for products with low technical complexity and a short life. With advances in modern science and technology, high-quality products with a long life are developed. Moreover, reliability analysis based on performance degradation data has become a new research focus [1–3]. Unlike traditional reliability analysis, performance-degradation-based reliability analysis considers the degradation process the main cause of failure. By measuring performance-degradation data, more life-cycle information can be obtained. In contemporary research, performance-degradation data are mainly fitted using the degradation-trajectory and stochastic-process models. The degradation of a product is affected by several factors, such as material characteristics and external shocks; therefore, degradation quantity shows a random variation throughout the product's life cycle. If a product is capable of self-recovery, the degradation process is likely nonmonotonic, which complicates the analytical process. Compared with the traditional degradation-trajectory model, using the Wiener [4,5] and gamma [6] processes for modeling is more advantageous and closer to the needs of the actual analysis.

Several studies discussed the self-recovery characteristics [7] of some materials [8] and electronic devices [9,10]; however, relatively few studies have conducted the reliability analysis of such products. Owing to the complexity of mechanical products and the various applications of new materials, the self-recovery phenomenon must be introduced into the reliability modeling of such products to assess their reliability. Qi analyzed the performance degradation and self-recovery process of a semiconductor laser [11,12]. Several studies

have verified the influence of the self-recovery process on system reliability via simulation analysis. In this study, the internal relationship between the shock interval and the self-recovery phenomenon, and the reliability modeling process with competitive failure are discussed. However, this model ignores that the degradation process of self-healing products is phased; phasing is related to changes in the working environment of the product, the characteristics of the material, and other factors.

Many scholars have examined the phenomenon of phased-product degradation. Dong set up two alarm thresholds related to the degradation amount in the degradation model [13]. The degradation process was divided into two different stages, and expressed with different mathematical models before and after the degradation reached the thresholds. Furthermore, after the system enters the second stage, the working time of products could lead to system failure. Tavangar [2] established a stochastic process reliability model, considering that at the junction of two degradation stages, degradation increases significantly. Gao [14] illustrated the universality of the two-stage degradation process with change points, considering lithium ion batteries, light-emitting diodes, and other electronic components as examples; they assumed that changes occurred at the degradation stage according to the cumulative degradation and the number of shocks. Another study [15] reported that the parameters of the soft failure threshold and the Wiener process drift change with changes in the degradation stage, and presented a simulation example verifying this phenomenon. Zhao [16] proposed a two-stage self-healing model based on cumulative and delta shocks, considering that self-recovery occurred only at the first stage of degradation. All the above-mentioned models had different definitions of phase change points. These articles, however, did not consider the fact that a product's working environment is alterable, and that such a change could cause the degradation process to enter the next stage.

Reliability modeling based on performance-degradation theory cannot be separated from the application of competitive failure; moreover, the interactions between shock and degradation processes cannot be ignored. Chang et al. [17] reported that reaching the performance-degradation threshold caused the decline of the hard-failure threshold. One study considered the delta shock in a reliability model, proposing that the shock causes degradation increment. An et al. [18] modeled the multidegradation process using the copula function, considering that the performance-degradation process was affected when the degradation was sufficiently large. Gao et al. [19] proposed two models; one considered that the arrival of effective shocks considerably increases degradation, and one considered that it accelerates the degradation process. These models have practical significance and cannot ignore the interactions between failure modes. However, few studies have considered the self-recovery of specific products.

Most contemporary stochastic process reliability models focus on the continuous degradation process, and few models discussed the impact of the shock process on performance degradation [20,21]. For example, semiconductor products are used in different working environments; thus, it is unreasonable to use only one degradation process model. The semiconductor industry has various unique production processes, so there are many restrictions for indices such as priority, production time, and storage time. The duration of the product inventory directly influences cumulative degradation before the working stage; thus, the initial degradation rate and inventory time should be analyzed. Therefore, to solve the above-mentioned problems, in this study, semiconductor products with self-recovery properties are the research object, meaning that the small failures of the semiconductor could be recovered through the intervention of a control unit or associated component to ensure that the semiconductor products continued to work normally, and to reduce the failure rate of products. Considering that there are two degradation processes during the service life of a product, a three-stage, stochastic-process, competitive-failure, reliability-evaluation model is proposed.



In this paper, the performance-degradation and self-recovery properties of products at the storage stage are considered, and the shock process was integrated into the study. A reliability-evaluation model of products in the whole life cycle is proposed that was verified with a simulation. The proposed model could describe the competitive failure process more accurately, and effectively prevent the product reliability from being over- or underrated due to the incompleteness of considering the product degradation features.

The article is organized as follows. The modeling description and basic derivation are presented in Section 2. The specific derivation process and results of this model are outlined in Section 3. In Section 4, a numerical example is presented to illustrate and verify the proposed model. Conclusions and the scope for future research are discussed in Section 5.

## 2. System Description and the Soft-Failure Process Analysis of Self-Recovery Products

### 2.1. System Description

The life cycle of a product comprises the storage period and working period. To ensure the universality of the model, the effects of the impact load and continuous degradation are considered together. Product performance degrades when the product is used continuously. Product failure occurs when degradation $X_s(t)$ reaches the certain degree of soft-failure threshold $D$ or the magnitude of external shock is greater than hard failure threshold $H$. Degradation volume accumulates during the storage period. The basic concepts assumed in this study are as follows:

1.  In a whole life cycle, the product has a storage phase $(0, T_1)$ and two work stages, $(T_1, T_2)$ and $(T_2, \infty)$. On the basis of the type of mathematical models used to describe the degradation processes, the three phases are the first and second Wiener processes, and the gamma process.
2.  Provided that the shock process follows a homogeneous Poisson process with intensity $\lambda$, the number of arrived shocks by time $t$ is $N(t) = n$. It was assumed that there was no shock in the first Wiener process.
3.  The degradation comprises continuous degradation and degradation increments caused by shocks. Self-recovery occurs when shock interval $B$ is greater than self-recovery threshold $\tau$. No shocks occur during the self-recovery process; therefore, there is no additional non-stationary interference to the degradation processes. In this situation, the gamma process is adequate to describe the third-phase degradation.
4.  The value range $[c_1, c_2]$ of cumulative degradation $c$ during the storage period is determined by analyzing the actual object, where $c_2$ denotes the upper limit of the cumulative degradation allowed in the storage stage, and $c_1$ is the minimal cumulative degradation allowed in the storage stage.
5.  The $i$ th shock follows a homogeneous Poisson process: $W_i \sim (\mu_W, \sigma_w^2)$. The corresponding degradation increment caused by $W_i$ is denoted as $Y_{0i} \sim N(\mu_3, \sigma_3^2)$.

The dependent competing failure processes of a product are shown in Figure 1.

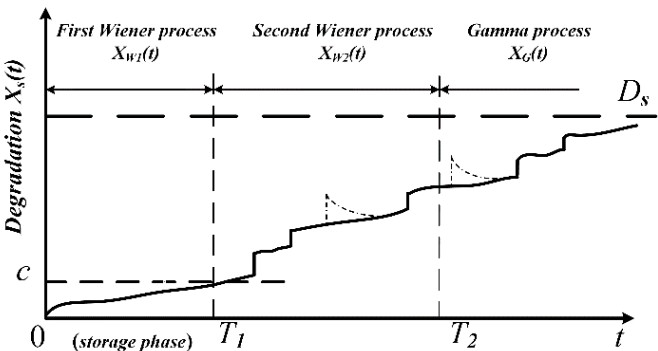

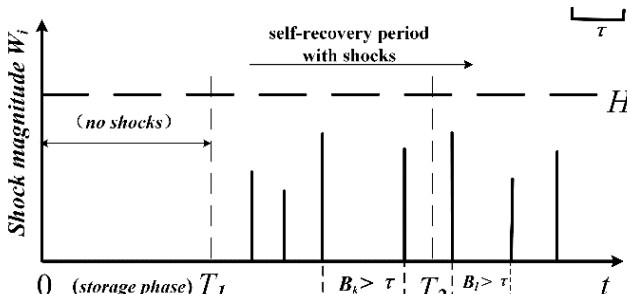

**Figure 1.** Dependent competing failure processes of a product.

*2.2. Stochastic Process Model*

The reliability model based on the Wiener process is expressed as follows:

$$X_W(t) = X_W(0) + \mu\Lambda(t;\theta_\Lambda) + \sigma B(\tau(t;\theta_\tau)) \tag{1}$$

where $X_W(0)$ is the initial degradation, and $\mu$ and $\sigma$ represent the drift and diffusion coefficients, respectively. Functions $\Lambda(t;\theta_\Lambda)$ and $\tau(t;\theta_\tau)$ describe the change in mean degradation and time transformation, respectively. The initial degradation was assumed to be $X_W(0) = 0$ and $\Lambda(t;\theta_\Lambda) = \tau(t;\theta_\tau) = t$. On the basis of the properties of the Wiener process, we obtained $X_W(t) \sim N(\mu t, \sigma^2 t)$.

The two Wiener processes mentioned above are expressed as follows:

$$X_{W1}(t) = \mu_1 t + \sigma_1 B(t) \tag{2}$$

$$X_{W2}(t) = \mu_2 t + \sigma_2 B(t) \tag{3}$$

Then, we obtained

$$X_{W1}(t) \sim N(\mu_1 t, \sigma_1^2 t) \tag{4}$$

$$X_{W2}(t) \sim N(\mu_2 t, \sigma_2^2 t) \tag{5}$$

When a soft failure occurs, the life of product $T_W$ is equal to the first passage time for the degradation threshold:

$$T_W = \inf\{t | X_W \geq D, t \geq 0\} \tag{6}$$

On the basis of the Wiener process, $X_{W1}(t) = \mu_1 + \sigma_1 B(t)$; $T_W$ follows an inverse Gaussian distribution. Let us assume that $T_{W1}$ represents the end time of the first Wiener

process. Then, we obtain the probability density function and the cumulative distribution function of $T_{W1}$ as follows:

$$f_{T_{W1}} = \frac{c}{\sqrt{2\pi\sigma_1^2 t^3}} \exp(-\frac{(c-\mu_1 t)^2}{2\sigma_1^2 t}) \tag{7}$$

$$F_{T_{W1}} = \phi(\frac{\mu_1 t - c}{\sigma_1\sqrt{t}}) + \exp(\frac{2\mu_1 c}{\sigma_1^2})\phi(-\frac{\mu_1 t + c}{\sigma_1\sqrt{t}}) \tag{8}$$

The gamma process is expressed as $G(t) \sim Ga(\beta(t), \eta(t))$, where $\beta(t)$ is the scale parameter, and $\eta(t)$ is the shape parameter. The probability density function of degradation $X_G(t)$ in this phase is expressed as follows:

$$f(X_G(t)|\eta(t), \beta(t)) = \frac{x^{\eta(t)-1}\beta(t)^{\eta(t)}e^{-\beta(t)x}}{\Gamma(\eta(t))}, x > 0 \tag{9}$$

$T_G$ is denoted as $T_G = \inf\{X_G(t) > D, t \geq 0\}$ in the gamma process. Let $\beta(t) = \beta$ and $\eta(t) = \alpha t$; then, the cumulative distribution function of Equation (9) can be expressed as follows:

$$\begin{aligned} F_{T_G}(t) &= P\{T_G \leq t\} = P\{X_G(t) \geq D\} = \int_D^\infty f(X_G(t)|\eta(t), \beta)dy \\ &= \int_D^\infty \frac{x^{\alpha t-1}\beta^{\alpha t}e^{-\beta x}}{\Gamma(\alpha t)}dx = \int_{D\beta}^\infty \frac{\mu^{\alpha t-1}e^{-\mu}}{\Gamma(\alpha t)}d\mu \\ &= \frac{\Gamma(\alpha t, D\beta)}{\Gamma(\alpha t)} \end{aligned} \tag{10}$$

where

$$\Gamma(a, b) = \int_b^\infty y^{a-1}e^{-y}dy \tag{11}$$

$$\Gamma(a) = \int_0^\infty y^{a-1}e^{-y}dy \tag{12}$$

On the basis of Equation (10), the reliability function of the gamma process can be expressed as follows:

$$R_G(t) = P\{T_G \geq t\} = 1 - F_{T_G}(t) = 1 - \frac{\Gamma(\alpha t, \beta D)}{\Gamma(\alpha t)} \tag{13}$$

### 2.3. Derivation of the Reliability Function for Self-Recovery Products

The cumulative distribution function of degradation increment $Y_{0i}$ caused by $W_i$ is expressed as $F_{Y_{0i}}(y) = p(Y_{0i} \leq y)$. The effective shock time by time $t$ is denoted as $N(t)$, and the total degradation increment is expressed as follows:

$$S_0(t) = \begin{cases} \sum_{i=1}^{N(t)} Y_{0i} & N(t) > 0 \\ 0 & N(t) = 0 \end{cases} \tag{14}$$

Due to the self-recovery property of the product, $S_0(t)$ was larger than the actual value. Let us assume that $S(t)$ represents the degradation increment considered in self-recovery, and the actual degradation increment of $W_i$ is $Y_i$. Then, to obtain the distribution function of $Y_i$, the cumulative distribution function is expressed as follows:

$$\begin{aligned} F_{Y_i}(y) &= p(Y_i \leq y) = 1 - p(Y_i > y) \\ &= 1 - [p(Y_{0i} > y|B_i \leq \tau) \times p(B_i \leq \tau) + p(Y_{0i} > y|B_i > \tau) \times p(B_i > \tau)] \end{aligned} \tag{15}$$

Self-recovery occurs when shock interval $B_i > \tau$, which means that $Y_{0i} = 0$. Thus, $p(Y_{0i} > y) = 0$. On the basis of the property of a homogeneous Poisson process, the shock interval $(\lambda)$ followed an exponential distribution. Thus, we obtained $p(B_i \leq \tau) = 1 - e^{-\lambda\tau}$, and Equation (15) could be expressed as follows:

$$
\begin{aligned}
F_{Y_i}(y) &= 1 - p(Y_{0i} > y | B_i \leq \tau) \times p(B_i \leq \tau) \\
&= p(Y_{0i} \leq y \,\big|\, B_i \leq \tau) \times p(B_i \leq \tau) = (1 - e^{-\lambda\tau}) F_{Y_{0i}}(y)
\end{aligned}
\tag{16}
$$

Moreover, Equation (16) is given as follows:

$$
p_{Y_i}(y) = (1 - e^{-\lambda\tau}) p_{Y_{0i}}(y) \tag{17}
$$

Let us assume that $K(t)$ represents the number of shocks that caused the degradation increment. On the basis of Equation (17), the ratio of $K(t)$ to all the shocks in $(0, t)$ can be derived as follows:

$$
p_1 = 1 - e^{-\lambda\tau} \tag{18}
$$

On the basis of Poisson's decomposition theorem and Equation (18), the self-recovery process $(\lambda e^{-\lambda\tau})$ and shock process $\left((1 - e^{-\lambda\tau})\lambda\right)$ could be determined, and both followed the Poisson process. The probability of $k_1$ times self-recovery occurring by time $t$ was calculated as follows:

$$
p\{K_1(t) = k_1\} = \frac{(e^{-\lambda\tau}\lambda t)^{k_1}}{k_1!} e^{-e^{-\lambda\tau}\lambda t} \tag{19}
$$

Considering the safety of the proposed model, the number of shocks causing the degradation increment can be expressed by rounding up the impact ration as follows:

$$
K(t) = p_1 N(t) + 1 \tag{20}
$$

On the basis of the definition of self-recovery, the product recovers from damage when $N(t) < 2$. The total degradation increment can be expressed as follows:

$$
S(t) = \begin{cases} \sum\limits_{i=1}^{K(t)} Y_i & N(t) \geq 2 \\ 0 & N(t) < 2 \end{cases} \tag{21}
$$

Let $N(t) = n$; then, when $n \geq 2$, we obtain

$$
S(t) = \sum_{i=1}^{K(t)} Y_i \sim N((np_1 + 1)\mu_3, (np_1 + 1)\sigma_3^2) \tag{22}
$$

Considering the self-recovery of the product, the total degradation by time $t$ is given as $X_s(t) = X(t) + S(t)$. Reliability in competing failure is expressed as follows:

$$
\begin{aligned}
R(t) &= \sum_{n=0}^{\infty} P(X(t) + S(t) < D, \bigcap_{i=1}^{N(t)} (W_i < H) | P(N(t) = n)) \times P(N(t) = n) \\
&= \sum_{n=0}^{\infty} R_{SF}(t | N(t) = n) \times p^n(W_i < H) \times p(N(t) = n)
\end{aligned}
\tag{23}
$$

## 3. Reliability Modeling for Self-Recovery Products in Whole Life Cycle

(1) When $t > T_2$, the product fully undergoes three phases. Degradation by time $t$ can be derived as follows:

$$
X(t) = X_{W1}(T_1) + X_{W2}(T_2 - T_1) + X_G(t - T_2) \tag{24}
$$

On the basis of Equations (2) and (3), Equation (24) can be expressed as follows:

$$X(t) = \mu_1 T_1 + \sigma_1 B(T_1) + \mu_2(T_2 - T_1) + \sigma_2 B(T_2 - T_1) + X_G(t - T_2) \tag{25}$$

Using Equations (21), (22), and (25), we can obtain the degradation by time $t$. The probability of no soft failure occurring can be expressed as follows:

$$
\begin{aligned}
&p_1(X(t) + S(t) < D) \\
&= p(\mu_1 T_1 + \sigma_1 B(T_1) + \mu_2(T_2 - T_1) + \sigma_2 B(T_2 - T_1) + X_G(t - T_2) + S(t) < D) \\
&= p(X_G(t - T_2) < D - (\mu_1 T_1 + \sigma_1 B(T_1) + \mu_2(T_2 - T_1) + \sigma_2 B(T_2 - T_1) + S(t)))
\end{aligned}
\tag{26}
$$

According to the classification shown in Equation (21), the right-hand side of Equation (26) is $A_{11}$ when $n \geq 2$, and $A_{12}$ when $n < 2$. Then, we obtain

$$A_{11} \sim N(D - \mu_1 T_1 - \mu_2(T_2 - T_1) - (np_1 + 1)\mu_3, \sigma_1^2 T_1 + \sigma_2^2(T_2 - T_1) + (np_1 + 1)\sigma_3^2) \tag{27}$$

$$A_{12} \sim N(D - \mu_1 T_1 - \mu_2(T_2 - T_1), \sigma_1^2 T_1 + \sigma_2^2(T_2 - T_1)) \tag{28}$$

On the basis of Equations (10) and (13), the probability of product survival in the soft-failure process is expressed as follows:

$$p_1(X(t) + S(t) < D) = \begin{cases} 1 - \dfrac{\Gamma(\alpha(t - T_2), \beta A_{11})}{\Gamma(\alpha(t - T_2))} & n \geq 2 \\[3mm] 1 - \dfrac{\Gamma(\alpha(t - T_2), \beta A_{12})}{\Gamma(\alpha(t - T_2))} & n < 2 \end{cases} \tag{29}$$

The probability of the two situations mentioned above is $p_{11}(t)$ and $p_{12}(t)$, respectively.

$T_1$ is introduced into the model by using Equation (7). Let us assume that $T_2$ follows a uniform distribution $U(T_1, t)$; then, the probability density function of $T_2$ is given as follows:

$$f_G(T_2) = \frac{1}{t - T_1} \tag{30}$$

On the basis of Section 2, the probability density function of cumulative degradation $c$ in the storage stage can be derived as follows:

$$f_C(c) = \frac{1}{c_2 - c_1} \frac{\Gamma(a + b)}{\Gamma(a)\Gamma(b)} \left(\frac{c - c_1}{c_2 - c_1}\right)^{a-1} \left(1 - \frac{c - c_1}{c_2 - c_1}\right)^{b-1} \tag{31}$$

In Equation (7), the probability density function of $T_1$ becomes

$$f_{T_1}^{W_1}(T_1) = \frac{c}{\sqrt{2\pi\sigma_1^2 t^3}} \exp\left(-\frac{(c - \mu_1 t)^2}{2\sigma_1^2 t}\right) \tag{32}$$

By combining Equations (27)–(32) and (25), when $n \geq 2$ and $n < 2$, reliability can be given as follows:

$$R_{SF11}(t) = \int_{c_1}^{c_2} \int_0^t \int_0^{T_2} p_{11}(t) \times f_{T_1}^{W_1}(T_1) \times f_G(T_2) \times f_C(c) dT_1 dT_2 dc \tag{33}$$

$$R_{SF12}(t) = \int_{c_1}^{c_2} \int_0^t \int_0^{T_2} p_{12}(t) \times f_{T_1}^{W_1}(T_1) \times f_G(T_2) \times f_C(c) dT_1 dT_2 dc \tag{34}$$

The first Wiener process had no impact arrival based on the hypothesis; thus, reliability in this situation can be expressed as follows:

$$
\begin{aligned}
R_1(t) = {} & \sum_{n=2}^{\infty} R_{SF11}(t|N(t-T_1)=n) \times p^n(W_i < H) \times p(N(t-T_1)=n) \\
& + \sum_{n=0}^{1} R_{SF12}(t|N(t-T_1)=n) \times p^n(W_i < H) \times p(N(t-T_1)=n)
\end{aligned}
\tag{35}
$$

where

$$
p(W_i < H) = \phi\left(\frac{H - \mu_W}{\sigma_W}\right)
\tag{36}
$$

$$
p(N(t-T_1)=n) = \frac{\exp(-\lambda(t-T_1))(\lambda(t-T_1))^n}{n!}
\tag{37}
$$

(2) When $T_1 < t < T_2$ the reliability model comprises the first and second Wiener processes. Total continuous degradation by time $t$ is given as follows:

$$
X(t) = X_{W1}(T_1) + X_{W2}(t-T_1) = \mu_1 T_1 + \sigma_1 B(T_1) + \mu_2(t-T_1) + \sigma_2 B(t-T_1)
\tag{38}
$$

The cumulative degradation at the end of the first Wiener process is considered a variable in interval $[c_1, c_2]$. The total degradation increment occurred in the second Wiener process. The probability of the product surviving from the soft-failure process was calculated as follows:

$$
\begin{aligned}
& p_2(X(t) + S(t) < D) \\
& = p(\mu_1 T_1 + \sigma_1 B(T_1) + \mu_2(t-T_1) + \sigma_2 B(t-T_1) + S(t) < D) \\
& = p(X_{W2}(t-T_1) < D - (c + S(t)))
\end{aligned}
\tag{39}
$$

The right-hand side of Equation (39) is marked as $A_2$ and given as follows:

$$
A_2 \sim N\left(D - c - (np_1 + 1)\mu_3, (np_1 + 1)\sigma_3^2\right)
\tag{40}
$$

On the basis of Equations (8) and (40), the cumulative distribution function by the end time of degradation is given as follows:

$$
\begin{aligned}
F_{T_{W2}}(t) & = P\{T_{W2} \le t\} = P\{X(t) \ge D\} \\
& = \phi\left(\frac{\mu_2(t-T_1) - A_2)}{\sigma_2 \sqrt{t-T_1}}\right) + \exp\left(\frac{2\mu_2 A_2}{\sigma_2^2}\right)\phi\left(-\frac{\mu_2(t-T_1) + A_2}{\sigma_2 \sqrt{(t-T_1)}}\right)
\end{aligned}
\tag{41}
$$

Then, Equation (41) becomes:

$$
\begin{aligned}
& p_2(X(t) + S(t) < D) \\
& = \begin{cases} \phi\left(\frac{A_2 - \mu_2(t-T_1))}{\sigma_2 \sqrt{t-T_1}}\right) - \exp\left(\frac{2\mu_2 A_2}{\sigma_2^2}\right)\phi\left(-\frac{\mu_2(t-T_1) + A_2}{\sigma_2 \sqrt{(t-T_1)}}\right) & n \ge 2 \\[2mm] \phi\left(\frac{(D-c) - \mu_2(t-T_1))}{\sigma_2 \sqrt{t-T_1}}\right) - \exp\left(\frac{2\mu_2(D-c)}{\sigma_2^2}\right)\phi\left(-\frac{\mu_2(t-T_1) + (D-c)}{\sigma_2 \sqrt{(t-T_1)}}\right) & n < 2 \end{cases}
\end{aligned}
\tag{42}
$$

The probability for the two situations mentioned in Equation (42) is labeled as $p_{21}(t)$ and $p_{22}(t)$, respectively. By combining Equations (7), (30) and (42), we obtain:

$$
R_{SF21}(t) = \int_{c_1}^{c_2} \int_0^t p_{21}(t) \times f_{T_1}^{W_1}(T_1) \times f_c(c) dT_1 dc
\tag{43}
$$

$$
R_{SF22}(t) = \int_0^{c_2} \int_0^t p_{22}(t) \times f_{T_1}^{W_1}(T_1) \times f_c(c) dT_1 dc
\tag{44}
$$



The overall reliability in $T_1 < t < T_2$ can be expressed as follows:

$$R_2(t) = \sum_{n=2}^{\infty} R_{SF21}(t \mid N(t - T_1) = n) \times p^n(W_i < H) \times p(N(t - T_1) = n)$$
$$+ \sum_{n=0}^{1} R_{SF2}(t \mid N(t - T_1) = n) \times p^n(W_i < H) \times p(N(t - T_1) = n) \tag{45}$$

(3) When $t < T_1$, the product was at the storage stage. On the basis of Equation (8), reliability is given as follows:

$$R_3(t) = \phi\left(\frac{D - \mu_1 t}{\sigma_1 \sqrt{t}}\right) - \exp\left(\frac{2\mu_1 D}{\sigma_1^2}\right)\phi\left(-\frac{\mu_1 t + D}{\sigma_1 \sqrt{t}}\right) \tag{46}$$

The running state of the product is one of the above cases; then, the overall reliability in the whole life cycle can be expressed as follows:

$$R(t) = \begin{cases} R_1(t) & t > T_2 \\ R_2(t) & T_1 < t \leq T_2 \\ R_3(t) & 0 < t \leq T_1 \end{cases} \tag{47}$$

## 4. Numerical Example

The resulting functions include variable bounds, and the integrand had variables such as $T_1$, $T_2$, and $c$; therefore, it was challenging to complete the calculation, so we propose an approximate solution. Moreover, we compared it with the result obtained by solving with the Monte Carlo method. The flowchart is presented in Figure 2. The initial setting of the parameters is an important step to validate the model. Semiconductor laser GB/T31359-2015 was tested according to its standard testing method at the ambient temperature of 2–23 °C and relative humidity of 5–55%. Failure occurred when the output optical power of the semiconductor laser was decreased to 55% of the initial optical power, that is, failure threshold $D = 55$. When the output optical power degradation had reached the failure threshold and could be recovered, the semiconductor laser did not fail. However, when the degradation of the output optical power of the semiconductor laser reached the failure threshold, but could not be recovered, the semiconductor laser failed. At the storage stage, the lower limit of natural degradation of semiconductor laser was $c_1 = 3$, and the upper limit was $c_2 = 5$. At the working stage, the high-temperature shock process of the semiconductor laser was a homogeneous Poisson process, the arrival rate of the high-temperature shock was $\lambda = 0.2$, the hard-failure threshold is $H = 5$, and the interval of self-recovery was $\tau = 3$. Parameter estimation showed that the drift coefficient and diffusion coefficient of the Wiener process were $\mu_1 = 0.5$, $\mu_2 = 2$, $\mu_3 = 0.5$, $\delta_1 = 0.1$, $\delta_2 = 1$, $\delta_3 = 0.1$. The shape and scale parameters were $\alpha = 3$ and $\beta = 1$, respectively. The time step used in this simulation was $\Delta t = 0.25$, and at each time point, $N = 100$ samples were processed. The average of the elemental values in the reliability array was taken as the reliability value of the respective point. MATLAB was used to fit the reliability curve with all reliability values in the range of [0, 50]. The black curves in Figures 3–8 show the reliability of each time point with the above-mentioned parameters. The curves with other colors are obtained results with different values of the key parameters indicating that the proposed model had good sensitivity.

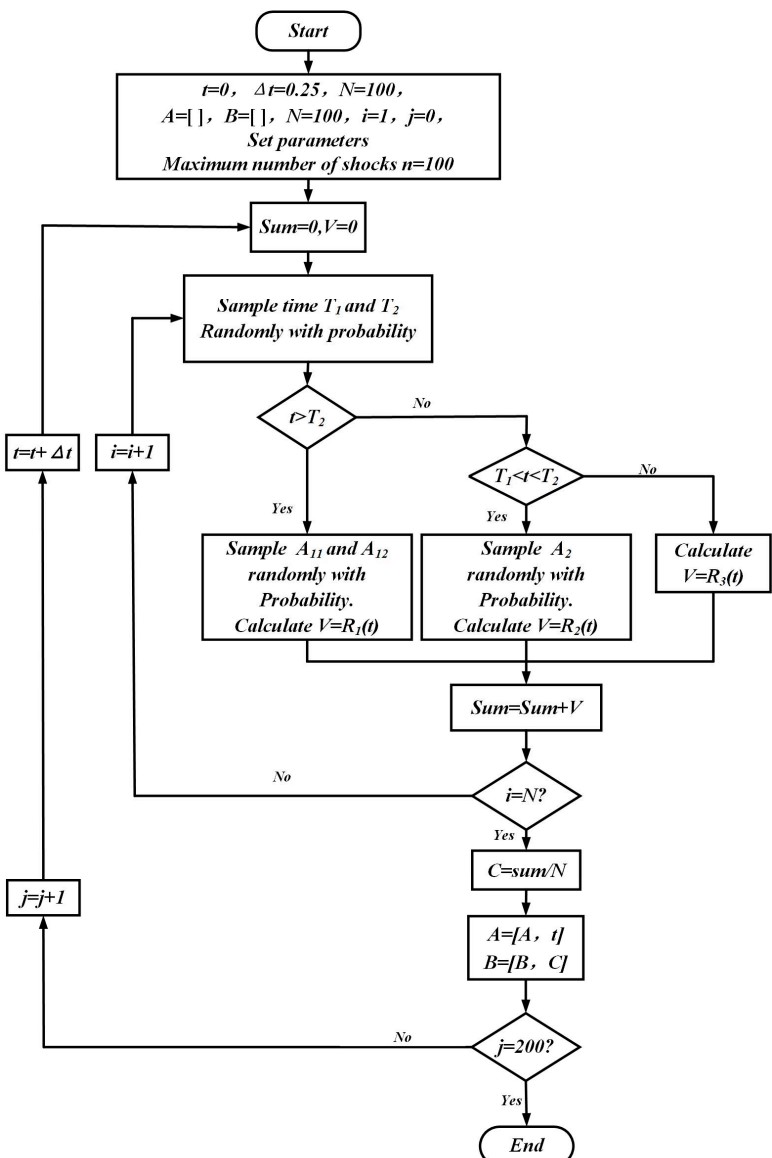

**Figure 2.** Flowchart of the reliability solving method.

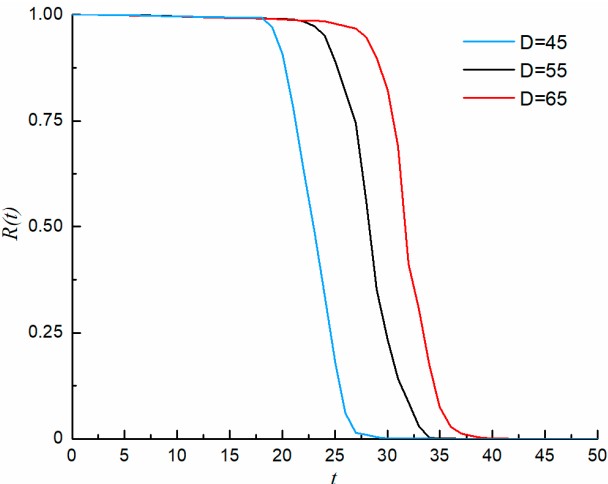

**Figure 3.** Reliability curves with varying soft-failure threshold.

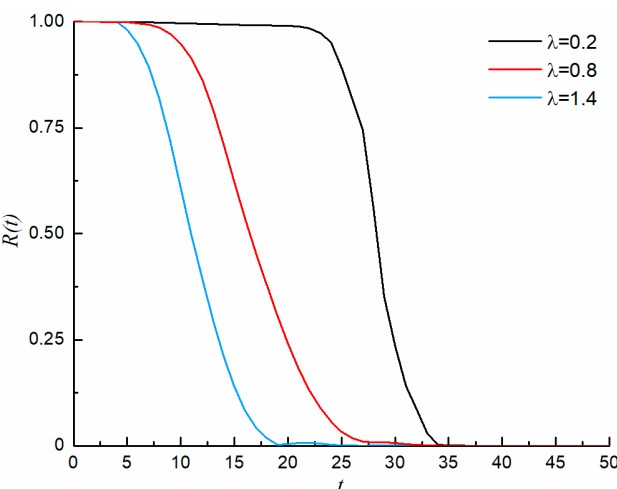

**Figure 4.** Reliability curves with varying arrival rates of the shock process.

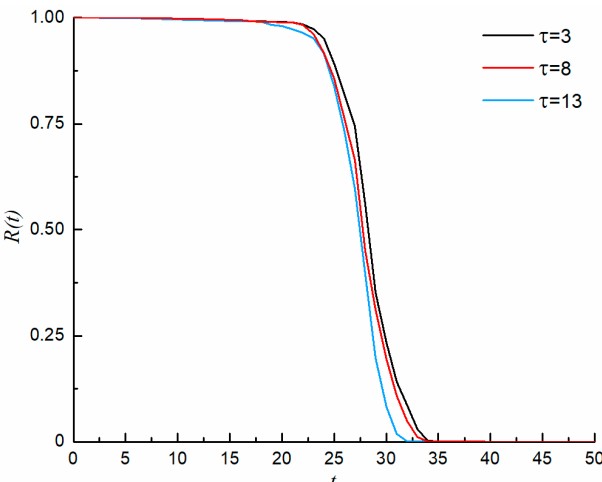

**Figure 5.** Reliability curves with varying interval thresholds of self-recovery.

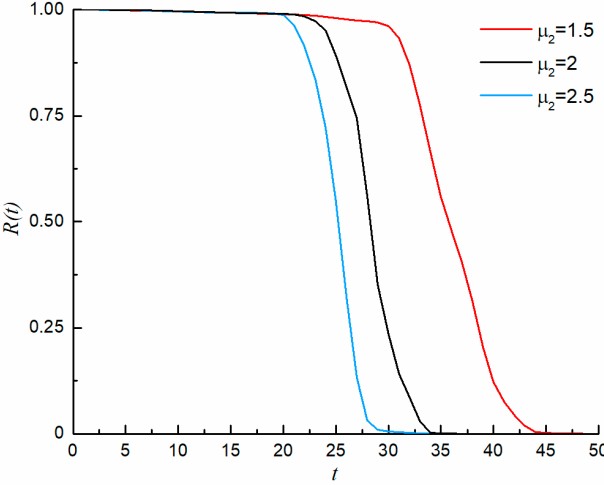

**Figure 6.** Reliability curves with varying drift parameters of the second Wiener process.

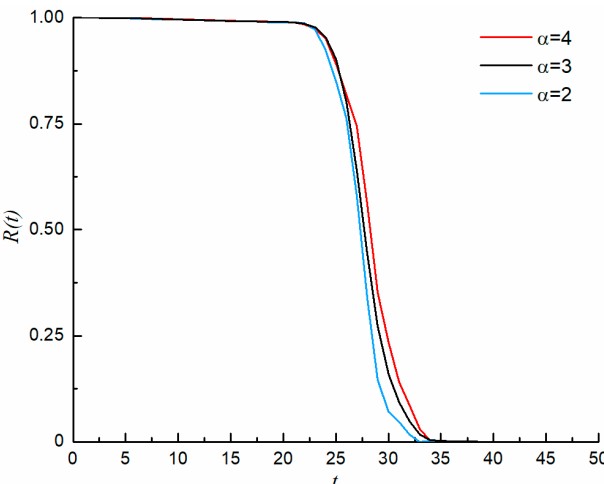

**Figure 7.** Reliability curves with varying shape coefficients of the gamma process.

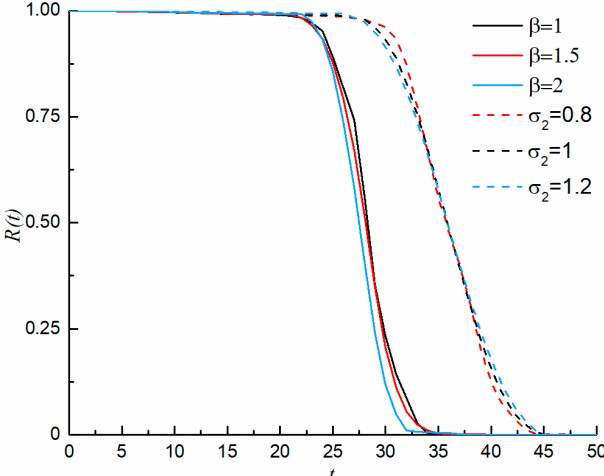

**Figure 8.** Reliability curves with varying diffusion parameters of the second Wiener process and varying scale parameters of the gamma process.

Figures 3 and 4 show the reliability curves with different arrival rates of the shock process and soft failure threshold, respectively. A larger arrival rate implies a lower curve. A higher arrival rate implies that more shocks arrived at the same time, and the damage caused by these shocks further accelerated the natural degradation process, which considerably decreased the product reliability. Similar changes occurred in reliabilities with a smaller soft-failure threshold. The working life of a product is longer when the initial value of the soft-failure threshold is high. Thus, an inaccurate set of soft-failure thresholds severely affects the final reliability and could produce a result that largely deviates from the actual circumstances. Therefore, the initial setting of soft-failure threshold $D$ requires a large amount of product data and much practical experience to ensure the actual effectiveness of the reliability model. The empirical threshold given from related studies, such as national standards, provides a reference basis for this paper.

Figure 5 demonstrates the influence of the self-recovery feature on the reliability of the considered products. In this model, the time interval between two shocks was set as the self-healing condition. Satisfying the condition is difficult when threshold $\tau$ increases, and reliability is lower when there are fewer cases of self-recovery.

The declining speed of the natural degradation process cuould be determined according to the drift parameters considered in the Wiener process. Larger parameter $\mu_2$ corresponds to a faster degradation rate. Figure 6 shows that a smaller value of parameter $\mu_2$ implied a slower decline of product reliability, resulting in a longer life.

Time $T_2$ was random, and the reliability of the product in $(T_1, t)$ was determined via the second Wiener and gamma processes. As shown in Figures 7 and 8, reliability increased with an increase in the shape parameter $\alpha$ or a decrease in scale parameter $\beta$. Equation (47) had many variables, and sampling was used in the calculation process. Thus, the computation fluctuated slightly; however, it did not affect the accuracy of the model.

On the basis of the original parameters above, parameter $\mu_2$ was 1.5. The dashed lines in Figure 8 comprise a set of curves obtained by changing the second Wiener process' diffusion parameter $\sigma_2$. Although the differences between the curves were not significant, an increase in reliability with the increase in diffusion parameter $\sigma_2$ was evident.

To verify the proposed model, the curves were compared with the results of the model with only the Wiener or gamma process. Self-recovery was also considered for the verification. Using Equations (43)–(46), the reliability obtained with the Wiener process could be expressed as follows:

$$
R_4(t) = \begin{cases} \sum\limits_{n=2}^{\infty} R_{SF21}(t|N(t-T_1)=n) \times p^n(W_i < H) \times p(N(t-T_1)=n) \\ + \sum\limits_{n=0}^{1} R_{SF22}(t|N(t-T_1)=n) \times p^n(W_i < H) \times p(N(t-T_1)=n) \qquad t > T_1 \\ \qquad \phi(\frac{D-\mu_1 t}{\sigma_1 \sqrt{t}}) - \exp(\frac{2\mu_1 D}{\sigma_1^2})\phi(-\frac{\mu_1 t + D}{\sigma_1 \sqrt{t}}) \qquad\qquad 0 < t \le T_1 \end{cases} \tag{48}
$$

Using the gamma process, Equation (26) becomes:

$$
\begin{aligned} p_2(X(t) + S(t) < D) &= p(\mu_1 T_1 + \sigma_1 B(T_1) + X_G(t - T_1) + S(t) < D) \\ &= p(X_G(t - T_1) < D - (\mu_1 T_1 + \sigma_1 B(T_1) + S(t))) \end{aligned} \tag{49}
$$

Similarly, when $n \ge 2$ and $n < 2$, the right-hand side of Equation (49) was $A_{51}$ and $A_{52}$, respectively. Then, the distribution is given as follows:

$$
A_{51} \sim N(D - \mu_1 T_1 - (np_1 + 1)\mu_3, \sigma_1^2 T_1 + (np_1 + 1)\sigma_3^2) \tag{50}
$$

$$
A_{52} \sim N(D - \mu_1 T_1, \sigma_1^2 T_1) \tag{51}
$$

As the model was described only with the gamma process, Equation (29) could be written as follows:

$$
p_2(X(t) + S(t) < D) = \begin{cases} 1 - \frac{\Gamma(\alpha(t-T_1), \beta A_{51})}{\Gamma(\alpha(t-T_1))} & n \ge 2 \\ 1 - \frac{\Gamma(\alpha(t-T_1), \beta A_{52})}{\Gamma(\alpha(t-T_1))} & n < 2 \end{cases} \tag{52}
$$

When $n \ge 2$ and $n < 2$, probabilities were $p_{21}(t)$ and $p_{22}(t)$, respectively. Then, Equations (33) and (34) are given as follows:

$$
R_{SF51}(t) = \int_{c_1}^{c_2} \int_0^t p_{21}(t) \times f_{T_1}^{W_1}(T_1) \times f_C(c) dT_1 dc \tag{53}
$$

$$
R_{SF52}(t) = \int_{c_1}^{c_2} \int_0^t p_{22}(t) \times f_{T_1}^{W_1}(T_1) \times f_C(c) dT_1 dc \tag{54}
$$

Reliability when $t > T_1$ is given as follows:

$$
\begin{aligned} R_{53}(t) = &\sum\limits_{n=2}^{\infty} R_{SF51}(t|N(t-T_1)=n) \times p^n(W_i < H) \times p(N(t-T_1)=n) \\ &+ \sum\limits_{n=0}^{1} R_{SF52}(t|N(t-T_1)=n) \times p^n(W_i < H) \times p(N(t-T_1)=n) \end{aligned} \tag{55}
$$

Using Equation (46), the reliability modeled with the gamma process is expressed as follows:

$$R_5(t) = \begin{cases} R_{53}(t) & t > T_1 \\ \phi(\frac{D-\mu_1 t}{\sigma_1 \sqrt{t}}) - \exp(\frac{2\mu_1 D}{\sigma_1^2})\phi(-\frac{\mu_1 t + D}{\sigma_1 \sqrt{t}}) & 0 < t \le T_1 \end{cases} \tag{56}$$

Figure 9 contains the curves obtained with Equations (48) and (56), and the simulation and analytical results based on the proposed model. The proposed model was closer to the actual situation, indicating its better accuracy; however, it was necessary to introduce self-recovery for the analysis of specific products.

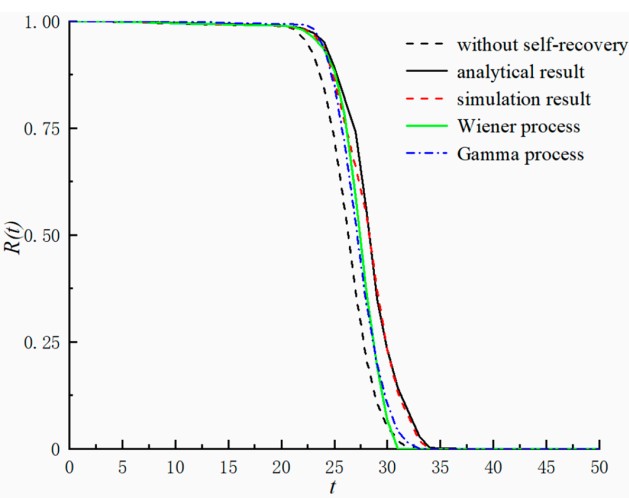

**Figure 9.** Reliability curves with varying models.

## 5. Conclusions

The proposed model in this paper considered all stages of the whole life cycle of the products and self-recovery properties, and improved the accuracy level of product reliability evaluation. As shown in Figure 9, compared with only one degradation model to evaluate product reliability, the accuracy of the proposed model was higher. Moreover, compared with the model that did not consider self-recovery properties, the accuracy of the proposed model was also higher. Therefore, when predicting the life or evaluating the reliability of products with self-recovery properties, the influence of self-recovery properties on reliability evaluation, and of performance degradation and threshold variation on product reliability should be considered in modeling. Through the analysis and proof of this paper, the following conclusions were drawn:

(1) If the influence of the degradation threshold variation on product reliability is not considered, the reliability of the product is overestimated, and the accident rate is increased.

(2) Ignoring the self-recovery properties of the product results in the reliability of the product being underestimated, which leads to increased costs of early operation and maintenance.

(3) With considering the impact of product degradation at the storage stage on product reliability, the proposed model had better accuracy.

In further research, in order to establish an improved reliability evaluation model with better accuracy that is more in line with actual working conditions, the influence of various stochastic processes and their combinations on the proposed model should be considered.

**Author Contributions:** Conceptualization, X.W. and B.W.; methodology, X.W.; software, X.W.; validation, X.W., B.W. and Z.H.; formal analysis, B.W.; investigation, X.W.; resources, X.W.; data curation, X.W.; writing—original draft preparation, X.W., B.W. and Y.N.; writing—review and editing, X.W. and Z.H.; visualization, Y.N.; supervision, X.W.; project administration, X.W. and B.W. All authors have read and agreed to the published version of the manuscript.

**Funding:** This research was funded by the Projects of Talents Recruitment of GDUPT (grand number 2020rc034), the State Key Laboratory of Precision Electronic Manufacturing Technology and Equipment (grand number JMDZ2021012), the Special Project in Key fields of the General Universities of Guangdong Province (grand number 2022ZDZX3013), the Natural Science Foundation of Hebei Province (grand number E2020501013), the Natural Science Foundation of Fujian Province (grand number 2022J01527), and CAST-BISEE2019-019.

**Institutional Review Board Statement:** Not applicable.

**Informed Consent Statement:** Not applicable.

**Data Availability Statement:** Not applicable.

**Conflicts of Interest:** The authors declare no conflict of interest.

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
