# Peer review of "Reliability Modeling of Products with Self-Recovery Features for Competing Failure Processes in Whole Life Cycle"

_applsci, doi:10.3390/app13084800_

Round 1

Reviewer 1 Report

Dear Authors,

I have reviewed manuscript entitled " Reliability modeling of products with self-recovery features for competing failure processes in whole life cycle" and I listed my comments below.

1. In Introduction section is sentence “With advances in modern science and technology, products with high quality and long life are being developed”. I am not sure that actually designers and manufacturers create and produce items (machines, cars, electronic items) for long life. Of course they are able to do it. In my opinion they design it rather for specified (required in a given situation) time and in my opinion not for long time because they want to sell next new items. Moreover, the requirements of people in the modern world are changing rapidly, so the producers want to supply new products in short time.

2. At the end of section 1 it is noted that the article is about semiconductor devices but in section 2 (System description) is mentioned the impact of bearing. Is there any connection? Maybe it should be more precisely explaining.

3. In the text it is not explain what is the physical nature of the self-recovery process in the case of the considered semiconductor products. It would be very helpfully for reader and helpful to understand the process and calculating model.

4. How in specific practical case one can obtain the coefficients values for presented Wiener process?

5. “probabilitydensityfunction” ???  before equation (7)

6. The last sentence in Conclusions gives information that reliability analysis without self-recovery process gives lower reliability than the actual value. One can say that of course it is less accurate but inaccuracy is safe because we think the reliability is lower than it really is.

7. The title of the journal indicates that articles showing the application of scientific solutions should be placed here. Thus, in my opinion, the authors could clearly indicate a specific case and conditions for which the presented solution may be applicable.

Author Response

Responses to the comments of Reviewer #1

  1. In Introduction section is sentence “With advances in modern science and technology, products with high quality and long life are being developed”. I am not sure that actually designers and manufacturers create and produce items (machines, cars, electronic items) for long life. Of course they are able to do it. In my opinion they design it rather for specified (required in a given situation) time and in my opinion not for long time because they want to sell next new items. Moreover, the requirements of people in the modern world are changing rapidly, so the producers want to supply new products in short time.

Response:

Thank you for pointing out an interesting question. With the development of the manufacturing industry, products do have the characteristics of high quality, long life, and high reliability compared to previous ones, resulting in lower product failure rates and lower maintenance costs. We don't consider the issue of sales market in the article.

  1. At the end of section 1 it is noted that the article is about semiconductor devices but in section 2 (System description) is mentioned the impact of bearing. Is there any connection? Maybe it should be more precisely explaining.

Response:

We were really sorry for our careless mistake. The original wrong content “impact bearing” is replaced by the correct content “impact load”.The object of the proposed model is semiconductor with self-recovery properties as we introduce in section 1.

  1. In the text it is not explain what is the physical nature of the self-recovery process in the case of the considered semiconductor products. It would be very helpfully for reader and helpful to understand the process and calculating model.

Response:

Thank you for your excellent suggestion. The self-recovery process of semiconductor products can be described as that the small failures of semiconductor can be recovered through the intervention of a control unit or associated component to ensure the semiconductor products continue to work normally, and finally to reduce the failure rate of products. The relevant content has been add at the section 1.

  1. How in specific practical case one can obtain the coefficients values for presented Wiener process?

Response:

Thank you for your precious question. In specific practical case, the coefficients values should be obtained from a large amount of the specific practical product data. It’s one of the direction for further research that improving the proposed model for specific practical application.

  1. “probabilitydensityfunction” ???  before equation (7)

Response:

Thank you for your careful check and feel sorry for our carelessness. The original content has been revised as the correct one “probability density function”.

  1. The last sentence in Conclusions gives information that reliability analysis without self-recovery process gives lower reliability than the actual value. One can say that of course it is less accurate but inaccuracy is safe because we think the reliability is lower than it really is.

Response:

Thank you for your critical question. The reliability model without self-recovery process do es show a low reliability, but the deviation is too large, which is leading to early maintenance and increasing costs. Therefore, self-recovery process should be considered.

  1. The title of the journal indicates that articles showing the application of scientific solutions should be placed here. Thus, in my opinion, the authors could clearly indicate a specific case and conditions for which the presented solution may be applicable.

Response:

Thank you for your precious suggestion. As we introduce in section 1, the semiconductor products with self-recovery properties is the applied object of the proposed model. In further research, we’ll try to obtain practical products data to improve the model for application at more specific products.

Reviewer 2 Report

It is not clear in which equations the authors interpret their own research as applied?

The conclusion does not contain the results presented in the article. The validity of the statements presented in the conclusion is quite controversial.

The authors unjustifiably limited the research to two laws of the distribution of random variables of a stochastic process.

The sequence of analytical statements is presented variegated. There is no initial hypothesis. The authors generally neglect constants of integration.

Author Response

Responses to the comments of Reviewer #2

  1. It is not clear in which equations the authors interpret their own research as applied?

Response:

Thank you for your critical question. In this article, the degradation processes of products are separated into one storage stage and two working stage. In addition to self-recovery properties of products, the shock process is also discussed. A reliability evaluation model for self-recovery products in whole life cycle is proposed.

  1. The conclusion does not contain the results presented in the article. The validity of the statements presented in the conclusion is quite controversial.

Response:

Thank you for pointing this problem for us. We have revised the conclusion according to your suggestion. As shown in figure 9 in the article, the proposed model is effective. Compared with reliability models constructed with only one stochastic process model, which leads an overrated reliability rate, the proposed three phases model has more accurate reliability, effectively ensuring the safety of the product during the service period. Moreover, as the result of comparison between the proposed model and the model without self-recovery, it’s obvious that the self-recovery feature is closely related to the shock process and must be considered in modeling, otherwise, the calculated reliability will be lower than the actual value, which is leading to early maintenance and increasing costs.

  1. The authors unjustifiably limited the research to two laws of the distribution of random variables of a stochastic process.

Response:

Thank your for your critical suggestion.It’s difficult to determine the distribution of random variables of a stochastic process because a large number of samples is required. Those two laws of the distribution of random variables of a stochastic process we researched are acquired from engineering experience and related literature, can support the validity of the research.

  1. The sequence of analytical statements is presented variegated. There is no initial hypothesis. The authors generally neglect constants of integration.

Response:

Thank you for pointing this problem. The constants of integration doesn't have any physical meaning here since there is no degradation at t=0, and no self-recovery and shock process at the initial stage, the reliability rate is 1.

Reviewer 3 Report

Reviewer’s Comments on applsci- 2247699

Reliability modeling of products with self-recovery features for competing failure processes in whole life cycle

This paper presents a degradation model that considers multiple phases, specifically a storage phase and two working phases. The model is based on a competing failure process with self-recovery. The modelling is implemented in a numerical example to demonstrate the applicability of the proposed approach. All in all, I consider that this is an interesting manuscript. I have the following comments:

1.     Although the authors present similar works to the proposed approach and discussed the shortcomings of these. I believe that the contributions of the proposed approach need to be more clearly stated in the introduction section. Please extend the discussion of the advantages of the proposed approach in comparison with the already presented discussion of related works.

2.     Why did the authors consider the gamma process for the third phase? It does not make sense to me that in the two first phases a non-stationary process is considered and in the last phase a stationary process is considered. If the considered degradation process is non-stationary, then the process should have a last phase with non-stationary behavior.

3.     Please extend the discuss about the effect of self-recovery in the behavior of the degradation process. Based on the previous comment, if the degradation process is stationary then the gamma process is adequate to model the process of the third phase, but the self-recovery effect may give a non-stationary behavior.

4.     I believe that the authors should present a comprehensive simulation study. Different combinations of stochastic processes may be considered to model the three considered phases of the life-cycle of products.

5.     Please extend the discussion of the details for the numerical example, specify the considered values for the critical degradation D.

6.     The authors are encouraged to provide insights for future research in the conclusion section. The proposed approach has several interesting opportunities to be extended.

Author Response

Responses to the comments of Reviewer #3

  1. Although the authors present similar works to the proposed approach and discussed the shortcomings of these. I believe that the contributions of the proposed approach need to be more clearly stated in the introduction section. Please extend the discussion of the advantages of the proposed approach in comparison with the already presented discussion of related works.

Response:

Thank you for your critical suggestion. We added relevant content in the introduction section, expounded the characteristics and advantages of the proposed model.

  1. Why did the authors consider the gamma process for the third phase? It does not make sense to me that in the two first phases a non-stationary process is considered and in the last phase a stationary process is considered. If the considered degradation process is non-stationary, then the process should have a last phase with non-stationary behavior.

Response:

Thank you for pointing this interesting question. Gamma process in the third phase is also a non-stationary process though its increment is stationary. Therefore, the last phase has a non-stationary behavior.

  1. Please extend the discuss about the effect of self-recovery in the behavior of the degradation process. Based on the previous comment, if the degradation process is stationary then the gamma process is adequate to model the process of the third phase, but the self-recovery effect may give a non-stationary behavior.

Response:

Thank your for your critical suggestion. We extended the discussion about the effect of self-recovery in the behavior of the degradation process. Self-recovery occurs when the shock interval is greater than the self-recovery threshold. No shocks occur during once self-recovery process, therefore, there’s no additional non-stationary interference to the degradation processes. In this situation, Gamma process is adequate to describe the the third phase degradation.

  1. I believe that the authors should present a comprehensive simulation study. Different combinations of stochastic processes may be considered to model the three considered phases of the life-cycle of products.

Response:

Thank you for your excellent suggestion. Three stochastic process are used in the degradation process in this paper. In order to determine the reliability of products more accurately, more types of stochastic processes and their combinations should indeed be considered in order to establish a more complete model. This is also the focus of follow-up research.

  1. Please extend the discussion of the details for the numerical example, specify the considered values for the critical degradation D.

Response:

Thank your for your critical suggestion. We extended the discussion in section 4. The the initial setting of soft failure threshold D requires a large amount of products data and practical experience to ensure the actual effectiveness of the reliability model. Empirical threshold given from related literature such as reference [16] provides a reference basis for this paper.

  1. The authors are encouraged to provide insights for future research in the conclusion section. The proposed approach has several interesting opportunities to be extended.

Response:

Thank you for your excellent suggestion. We added the content about the further research of our study. In order to establish a more improved reliability evaluation model with higher accuracy and more in line with actual working conditions, the influence of various types of of stochastic processes and their combinations on the proposed model should be considered. Moreover, Investigating obtaining a large number of accumulated data of practical products to determine more precise parameters and distributions of random variables to improve the proposed model might prove important.

Round 2

Reviewer 2 Report

The authors provided abstract and philosophical explanations for the reviewer's comments. None of the explanations contains the validity of the mathematical approach of summarizing research results.

Author Response

Thanks for your valuable comments, and hereby make the following explanations for the review comments:

1) The conclusion is modified and the main points of the conclusion are clarified.

2) In view of conclusion 1, the variation of product reliability under variable threshold is shown in Figure 3& 4. Soft failure has an impact on hard failure threshold, and hard failure in turn has an impact on soft failure threshold. In this paper,  by changing the threshold value, it is shown that the reliability of the product will be overestimated and the accident rate will be increased without considering the impact of the variation of the threshold value on the product reliability.

3) In view of conclusion 2, the influence of self-recovery propertieson product reliability is discussed in Section 3.1, 3.2 and 4. When the shock interval is greater than τ, the product can recover to the state before damage, that is, self-recovery. Ignoring the self-recovery properties of the product leads to that the reliability of the product is underestimated and the cost of operation and maintenance is increased.

3) In view of conclusion 3, the product degradation process during the storagestage is discussed in Section 3.1, and a mathematical model is established. With considering that the reliability of products will be reduced after the storage stage, the model has a better accuracy of product reliability evaluation.

Reviewer 3 Report

The authors have provided responses to the provided comments, although I'm still have some comments:

- What do the authors imply in the response for the second comment "gamma process in the third phase is also a non-stationary though its increment is stationary"? Please discuss how is this possible. Furthermore, please complement the corresponding content in the manuscript, the response should be considered in the improved version of the manuscript. In this sense, please complement Figure 1 and improve its quality.

-The numerical example, is still not clear to me, the authors provided a reference [16], where the parameters were considered, but the description of the scenarios is not completely discussed. Please organize the contents of the section and clearly discuss all the details of the example.

Author Response

Comments:

The authors have provided responses to the provided comments, although I'm still have some comments:

- What do the authors imply in the response for the second comment "gamma process in the third phase is also a non-stationary though its increment is stationary"? Please discuss how is this possible. Furthermore, please complement the corresponding content in the manuscript, the response should be considered in the improved version of the manuscript. In this sense, please complement Figure 1 and improve its quality.

-The numerical example, is still not clear to me, the authors provided a reference [16], where the parameters were considered, but the description of the scenarios is not completely discussed. Please organize the contents of the section and clearly discuss all the details of the example.

Responses:

Thanks for your valuable comments, and hereby make the following explanations for the review comments:

Question 1: The first stage is the storage stage, including temperature shocks, humidity shocks and unpredictable vibration shocks, therefore a non-stationary random process is adopted to describe this stage. The second stage is the initial working stage of the product. Due to the randomness of the product's own materials and the instability of the working environment, the initial working state of the product is also non-stationary, and its reliability follows the initial stage of the bathtub curve, so the non-stationary random process is also adopted to describe it. The third stage is the normal working stage of the product, and the shape parameter is a linear function, so the stationary gamma process is used to describe the third stage.

If the three stages of work are described by the same random process, the model and method adopted in the whole paper will be too simple. The degradation process is complex, and the more different types of models can be used to describe it, the more accurate it will be. Therefore, this is also the reason for the adoption of gamma process description in the third stage.

Question2: According to the standard testing method of semiconductor laser GB/T31359-2015, the semiconductor laser is tested at the ambient temperature of 23℃ ~ 2℃ and the relative humidity of 55% ~ 5%. Failure occurs when the output optical power of the semiconductor laser decreases to 55% of the initial optical power, that is, the failure threshold D=55. When the output optical power degradation reaches the failure threshold and can be recovered, the semiconductor laser does not fail. However, when the degradation of the output optical power of the semiconductor laser reaches the failure threshold but cannot be recovered, the semiconductor laser fails. In the storage stage, the lower limit of natural degradation of semiconductor laser is c1 = 3, and the upper limit is c2 = 5. In the working stage, the high temperature shock process of the semiconductor laser is a homogeneous Poisson process, and the arrival rate of the high temperature shock is λ = 0.2, the hard failure threshold is H = 5, and the interval of self-recovery is τ = 3. By parameters estimation, it can be obtained that the drift coefficient and diffusion coefficient of the Wiener process are μ1 = 0.5, μ2 = 2, μ3 = 0.5, δ1 = 0.1, δ2 = 1, δ3 = 0.1. The shape parameter and scale parameter are α=3, β=1, respectively. In this paper, step size Δt = 0.25 is selected and N = 100 times of sampling was conducted at each time point, and the average number of elements in the obtained reliability group is taken as the reliability value of this point.

Round 3

Reviewer 2 Report

Ok.

Reviewer 3 Report

The authors have considered the provided comments to improve the manuscript. I believe this version can be considered for publication.